# SUBP: Soft Uniform Block Pruning for $1\times$N Sparse CNNs Multithreading Acceleration

**Jingyang Xiang**[1]     **Siqi Li**[1]    **Jun Chen**[1]    **Shipeng Bai**[1]
**Yukai Ma**[1]    **Guang Dai**[2,3]    **Yong Liu**[1]*

[1]APRIL Lab, Zhejiang University, Hangzhou, China
[2]State Grid Corporation of China
[3]SGIT AI Lab, Shaanxi, China

{jingyangxiang,lsq4747,junc,shipengbai,yukaima}@zju.edu.cn
guang.gdai@gmail.com,yongliu@iipc.zju.edu.cn

## Abstract

The study of sparsity in Convolutional Neural Networks (CNNs) has become widespread to compress and accelerate models in environments with limited resources. By constraining N consecutive weights along the output channel to be group-wise non-zero, the recent network with $1\times$N sparsity has received tremendous popularity for its three outstanding advantages: 1) A large amount of storage space saving by a *Block Sparse Row* matrix. 2) Excellent performance at a high sparsity. 3) Significant speedups on CPUs with Advanced Vector Extensions. Recent work requires selecting and fine-tuning $1\times$N sparse weights based on dense pre-trained weights, leading to the problems such as expensive training cost and memory access, sub-optimal model quality, as well as unbalanced workload across threads (different sparsity across output channels). To overcome them, this paper proposes a novel *Soft Uniform Block Pruning* (SUBP) approach to train a uniform $1\times$N sparse structured network from scratch. Specifically, our approach tends to repeatedly allow pruned blocks to regrow to the network based on block angular redundancy and importance sampling in a uniform manner throughout the training process. It not only makes the model less dependent on pre-training, reduces the model redundancy and the risk of pruning the important blocks permanently but also achieves balanced workload. Empirically, on ImageNet, comprehensive experiments across various CNN architectures show that our SUBP consistently outperforms existing $1\times$N and structured sparsity methods based on pre-trained models or training from scratch. Source codes and models are available at `https://github.com/JingyangXiang/SUBP`.

## 1   Introduction

In recent years, convolutional neural networks (CNNs) have achieved great success in image classification [56, 22], object detection [21, 16], semantic segmentation [17, 4] and other fields. The remarkable performance owes to the deeper and wider architectures, also leading to the prohibitively expensive computational cost and colossal memory footprint. Although some efficient architectures are proposed, such as residual connection [22], inception module [58], etc., it is still difficult to deploy the state-of-the-art CNNs on the available CPUs embedded devices with limited resource. Therefore, the network pruning is emerging by pruning the redundancy in CNNs. Due to the appealing perfor-

---

*Corresponding author

37th Conference on Neural Information Processing Systems (NeurIPS 2023).

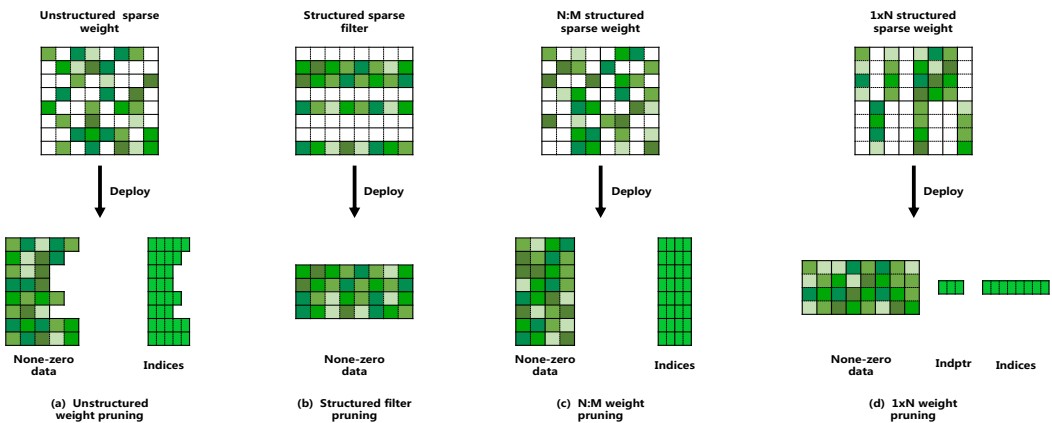

Figure 1: Four mainstream types of pruning in the literature. (a) Unstructured weight pruning removes individual weights at arbitrary locations. (b) Structured filter pruning removes entire convolutional filters. (c) N:M weight pruning (2:4 case) requires at most N out of M consecutive weights along input channels to be non-zero. (d)1×N weight pruning (1×4 case) constrains N consecutive weights along the output channel to be group-wise non-zero.

mance, it has received extensive attention from industry and academia. It is necessary to maintain the model with small size, low memory footprint, low computational cost and high inference speed.

According to the pruning granularity, most existing works about network pruning mainly focus on weight pruning [10, 19, 34] and filter pruning [64, 40, 25, 24, 46]. Weight pruning deletes the weight of filters directly, which may always result in the unstructured sparsity of filters. The expensive memory accesses is also less efficient in saving memory usage and computational cost, since the unstructured model needs a large number of indices to record the positions of the reserved weights and can't take full advantage of *Advanced Vector Extensions* (AVX), *Single Instruction Multiple Data* (SIMD) and poorly utilizes memory caches [63]. In contrast, filter pruning tends to prune at the level of filter or even layer. Since filter pruning still preserves the original convolution structure, it enables the model with structured sparsity and more efficient memory usage. Therefore, filter pruning can take full advantage of the high-performance computing library i.e. *Basic Linear Algebra Subprogram* (BLAS) to achieve apparent acceleration and is more advocated in accelerating the networks. More recently, the development of hardware and operators has given rise to new pruning patterns. The most famous work is N:M fine-grained pruning [55], where N out of M weights are zeros for every continuous M weights along input channels. It also can be seen as a special pattern of weight pruning. Currently, this pattern achieves acceleration only in the case of 2:4. However, it is impossible to be utilized on other types of devices since the instructions of sparse *Matrix Multiply-Accumulate* (MMA) are specially designed for NVIDIA Ampere Core [55]. Not to mention, in realistic deployment scenarios, GPUs on mobile and embedded devices are not always accessible, and such GPUs are more difficult to be satisfied.

Above all, how to retain the performance and achieve realistic acceleration on mobile CPUs becomes a challengeable but valuable problem. In order to solve this issue, Lin *et al.* [65] proposed a novel pattern of 1×N weight pruning with its merits in realizing both high-performing accuracy and apparent CPUs acceleration for practical model deployment. The 1×N pruning pattern provides an intermediate granular level for network pruning, which is coarser as compared to the fine-grained weight but finer as compared to the coarse-grained filter. Fig. 1(d) shows an example of 1×N pruning pattern that satisfies N=4, the core distinction lies in that 1×N pruning consists of N consecutive weights along the output channel to be group-wise non-zero. These consecutive weights can be stored continuously in the memory cache and the convolution with the inputs can proceed using a block-wise vectorized operation in parallel thanks to AVX and SIMD. The indices memory of the weight positions can also benefit from *Block Sparse Row* (BSR) matrix and be greatly saved.

However, Lin *et al.* [65] permanently pruned blocks based on "smaller-norm-less-important" criterion in a non-uniform manner. On the one hand, it reduced the capacity of original model and thus harmed the performance. On the other hand, it left the blocks redundancy untouched and always caused unbalanced workload across threads. What's more, it still followed a traditional pre-training, pruning

and fine-tuning (PPF) pipeline, which depended on pre-trained model and still suffered from the expensive pre-training burden.

To address the above-mentioned limitations, we propose a novel block pruning approach named *Soft Uniform **B**lock **P**runing* (SUBP) to obtain high accuracy sub-model without PPF pipeline. In contrast to the traditional pruning approaches that non-uniformly and permanently remove blocks, we prune and regrow the blocks uniformly via importance sampling, which allows the pruned blocks to be recovered and balanced workload across threads. From intra-layer's perspective, we propose a new *Block **P**runing* criterion by taking *Angular **R**edundancy* (BPAR) into account. BPAR identifies the angular redundancy between blocks, so we can prune blocks with redundancy, rather than those with "relatively less" importance. With only one training procedure from scratch, our obtained sub-models yield better accuracy than the previous methods under the similar FLOPs constraints.

To sum up, the contributions of this paper are highlighted as follows:

- We propose a novel block pruning approach named *Soft **U**niform **B**lock **P**runing* (SUBP) with three appealing characteristics: (1) a periodic block pruning and regrowing technique via importance sampling, (2) a pruning criterion based on angular redundancy across blocks, and (3) a uniform $1 \times N$ sparse pattern for multithreading acceleration.

- Our approach trains a uniform $1 \times N$ sparse CNNs from scratch, effectively reducing the training cost and achieving better inference latencies in multithreading scenarios, since it circumvents the expensive PPF pipeline and balances workload across threads.

- Extensive experiments on the large ImageNet dataset have demonstrated the effectiveness of our SUBP under different FLOPs constraints. SUBP obtains consistent accuracy improvement across various N and networks, achieving a better trade-off between accuracy and inference latencies. For example, ResNet50 ($1 \times 16$) model yields $4 \times$ FLOPs reduction while still achieving 76.3% top-1 accuracy and suppressing the previous results.

## 2 Related Work

Fully connected operators are widely used in various types of neural networks, and they can be mathematically represented by one matrix multiplication. The natural idea is that all elements of a matrix are not equally important. Removing unimportant elements from the fully connected operators not only reduces the size and the amount of computation, but also potentially improves the generalization performance of the model. Weight pruning (Fig. 1(a)) and filter pruning (Fig. 1(b)) are traditional pruning methods. There are also some special pruning methods that require a high degree of integration with hardware or operators as shown in Fig. 1(c,d). In what follows, we will briefly review some related works.

**Weight Pruning.** Weight pruning is one of the most widely studied model pruning methods, which removes individual weights at any position of the network. The study of weight pruning could date back to optimal brain damage [33] and optimal surgeon [20], which prune weights based on the Hessian within the loss function. Previous studies removed unimportant weights by using gradient [34], momentum [10], magnitude [19], *etc*. Han *et al.* [19] proposed to discard the small weights whose values are below the threshold through an iterative method. Recent some works also payed attention to unstructured sparsity in an adaptive training manner. Ding *et al.* [10] gradually zeroed out the redundant weights by categorizing weights into two parts and updating them according to different rules. Frankle *et al.* [13] presented an algorithm to identify subnetworks that were capable of training effectively. Although weight pruning can maintain most of the accuracy of the model with high sparsity, it is difficult to leverage the existing high-efficiency BLAS libraries in practice to accelerate on general hardware due to its irregular weight distribution. Therefore, weight sparsity pruning is rarely used in practical applications.

**Filter Pruning.** Filter pruning gains achieve noticeable speedup on general hardware after pruning. The filter importance and the pruned network structure are the two most important issues widely studied by researchers. Typical works solve the former issue by devising a certain criterion to measure the importance of filters, including output activation [64], scale factor amplitude of Batch Normalization layer [46], ranks of feature map [40], norm or geometric median of filters [24, 25], *etc.*. To be specific, he *et al.* [25] leveraged geometric median to prune filters with redundancy. As for the latter, most works are based on rules of thumb [7, 24], or use evolutionary algorithms [41],

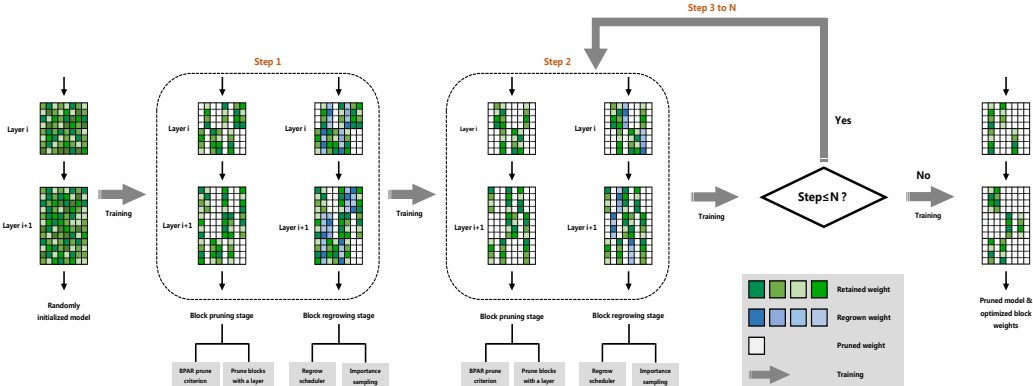

Figure 2: An illustration of our SUBP method, which optimizes the weight values and regrows the removed blocks in one training pass from scratch jointly. In SUBP, both retained and regrown blocks are active, participating in the training iterations. After the last iteration, SUBP exports the pruned model and optimized block weights.

reinforcement learning [26], meta learning [45] and other methods [9, 43] to predict the layer-wise sparsity. For instance, he *et al.* [26] applied reinforcement learning to compression and acceleration models on mobile devices automatically. However, filter pruning removes entire convolution filters, which may damage the information learned by the network and cause serious accuracy degradation with a high pruning rate. Therefore, filter pruning is still difficult to apply in practical tasks.

**Special Pruning.** In order to achieve the maximum balance between accuracy and model performance, researchers have proposed various special pruning types [49, 50, 3, 12]. These convolution modes often require the combination of special operators or hardwares. Supported by the NVIDIA Ampere Core, N:M sparsity has gained tremendous attention due to its attractive storage and computation efficiency. Asit *et al.* [50] and Pool *et al.* [53] followed a traditional PPF pipeline to implement N:M sparsity. Although retraining can improve the accuracy of the N:M sparsity network, the pre-training-based approach still incurs expensive training costs, which hinders the deployment of N:M sparsity techniques. To address this, zhou *et al.* [66] proposed a Sparse-refined straight-through estimator (SR-STE) to train N:M sparsity network from scratch. Zhang *et al.* [65] characterized N:M sparsity as a combinatorial problem and assigned each combination a learnable score to obtain the best subset. Apart from these, researchers have also turned their attention to $1\times$N sparse networks, which can be accelerated on CPUs. Lin *et al.* [42] proposed $1\times$N pruning pattern for CNNs firstly, which achieved better accuracy and speed trade-off than both weight and filter pruning.

## 3 Methodology

### 3.1 Uniform 1×N Block Pruning

We start by introducing symbols and notations in this subsection. Without losing generality, we assume that a CNN has $L$ layers. We parameterize the tensor connection of CNN with $\{W^i \in \mathbb{R}^{C_{i+1}\times C_i\times K_i^h\times K_i^w}|1 \le i \le L\}$, where $C_i$, $C_{i+1}$, $K_i^h$ and $K_i^w$ represent the numbers of input channels, output channels, kernel height and kernel width of $i$-th layer, respectively.

A $1\times$N pruning pattern partitions the whole $W^i$ into a collection of small blocks. Considering the $W^i$ in the $i$-th layer, the $1\times$N pruning pattern can be achieved by partitioning $W^i$ into a collection of $C_i$ col-groups and then further partitioning $N_i$ into a collection of $\frac{C_{i+1}}{N}$ row-groups. Consequently, each block is a $1\times$N matrix along the input channel, which includes N consecutive output kernel with the same input channel index. We denote the matrix block set as $\{B_{j,k}^i \in \mathbb{R}^{N\times 1\times K_i^h\times K_i^w}|1 \le i \le L, 0 \le j \le \frac{C_{i+1}}{N} - 1, 0 \le k \le C_i - 1\}$, to stand for the $W_{j\cdot N:(j+1)\cdot N,k,:,:}^i$. Based on this partition pattern, the basic pruning granularity of $1\times$N sparsity falls into these blocks. According to network pruning which can be implemented by imposing a mask $M^i$ upon $W^i$, we introduce $\{M_{j,k}^i \in \{0,1\}|1 \le i \le L, 0 \le j \le \frac{C_{i+1}}{N} - 1, 0 \le k \le C_i - 1\}$, to define the objective function of

pre-existing work:

$$\arg\max_{M^i} \sum_{i=1}^{L} \mathcal{F}(\{B_{j,k}^i \cdot M_{j,k}^i | 0 \leq j \leq \frac{C_{i+1}}{N} - 1, 0 \leq k \leq C_i - 1\}), \ s.t. \frac{\|M_{:,:}^i\|_0}{K} = 1 - p \quad (1)$$

where $\mathcal{F}(\cdot)$ measures the importance of its input, $K = \frac{C_{i+1}}{N} \cdot C_i$ and $p$ is the expected prune rate for the model.

To reduce the latency of CNNs, multi-core devices are often adopted in most practical scenarios. On a single-core CPU, the most important aspect is the continuity and locality of memory access and calculation in order to fully utilize the cache and vectorization unit. However, nearly all modern processors have multiple cores, and the computation can benefit from multithreading parallelism. Even though the continuity and locality of memory access and calculation are also important to multi-core CPU, the most important aspect of CPU multi-core computing is to achieve balanced workload to fully utilize the computing power of each core and improve computational efficiency. As can be seen from Fig. 3, if the sparsity among different output channel blocks keeps the same, the workload across different threads will be the same.

However, it is easy to know that the conditions in Eq. (1) often lead to different sparsity levels among different output channel blocks, e.g., $\|M_{1,:}^i\|_0 = 1$ and $\|M_{2,:}^i\|_0 = C_i - 1$, which will cause workload imbalance [60] among threads, degrade performance and waste resources during multicore execution. Therefore, to overcome these issues, we propose a uniform $1{\times}N$ block pruning type on the basis of the analysis in above, which is equivalent to constraining the sparsity levels among different output channel blocks to be the same. In another word, the uniform $1{\times}N$ block pruning is a special case of $1{\times}N$ block pruning. Therefore, we define the objective function of uniform $1{\times}N$ block pruning as:

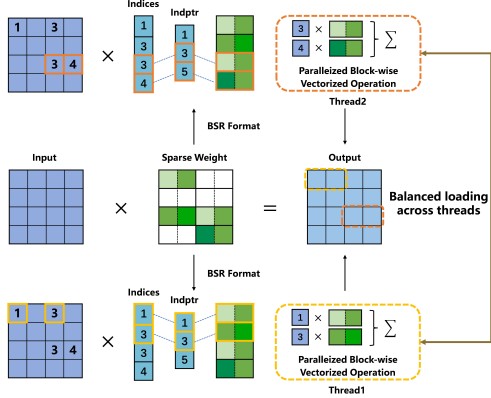

Figure 3: Balanced workload across threads.

$$\arg\max_{M^i} \sum_{i=1}^{L} \sum_{j=0}^{\frac{C_{i+1}}{N}-1} \mathcal{F}(\{B_{j,k}^i \cdot M_{j,k}^i | 0 \leq k \leq C_i - 1\}), \ s.t. \frac{\|M_{j,:}^i\|_0}{C_i} = 1 - p \quad (2)$$

## 3.2 Block Pruning via Angular Redundancy

To get rid of the constraints in the norm-based criterion, we propose a new block pruning criterion by taking block angular redundancy into account and name it as BPAR here. The central idea of angular redundancy is as follows: given a non-zero vector $X \in \mathbb{R}^{N \times M}$ and a non-zero vector $W \in \mathbb{R}^{N \times 1}$, we have

$$\begin{cases} W_1^T X = \alpha W_2^T X, & < W_1, W_2 >= 0 \\ W_1^T X = -\alpha W_2^T X, & < W_1, W_2 >= \pi \end{cases}, \ \alpha = \frac{\|W_1\|_2}{\|W_2\|_2} \quad (3)$$

where $< \cdot, \cdot >$ denotes the angle between two vectors. In particular, we define the $< \mathbf{0}, \cdot >= 0$. According to the above analyses, even though a $1{\times}N$ block may be relatively important, this block still has an angular redundancy if there exists another block with a similar orientation angle. Similarly, a relatively less important $1{\times}N$ block can also map the vectors to a meaningful space and extract valuable information if its orientation angle is very different from others.

In order to simplify the explanations, firstly, we vectorize the representation of $B_{j,k}^i \in \mathbb{R}^{N \times 1 \times K_i^h \times K_i^w}$ as $\Omega_{j,k}^i \in \mathbb{R}^{N * 1 * K_i^h * K_i^w}$. Then, we compute the importance score $S$ of our block as:

$$S_{j,k}^i = \frac{\|\Omega_{j,k}^i\|_1}{\sum_{m=0}^{C_i-1} \|\Omega_{j,m}^i\|_1} - \lambda \frac{\sum_{m=0}^{C_i-1} \left| CosineSim(\Omega_{j,k}^i, \Omega_{j,m}^i) \right|}{\sum_{n=0}^{C_i-1} \sum_{m=0}^{C_i-1} \left| CosineSim(\Omega_{j,n}^i, \Omega_{j,m}^i) \right|}, \quad (4)$$

where $\lambda$ is a balanced hyper-parameter and $CosineSim$ denotes cosine distance between two vectors. Based on Eq. (4), it is obvious that unlike previous criterion solely based on norm, our approach might retain some blocks with smaller norm but larger angular differences in comparison with other blocks. Simultaneously, it may discard the blocks with larger norm but smaller angular differences.

### 3.3 Soft Uniform Block Pruning

Most of the previous pruning [27, 40, 42] works followed a traditional PPF pipeline, which compressed CNNs in a hard manner. However, once filters or weights are pruned, they can not be recovered, which limits the model capacity and leads to unacceptable accuracy degradation.

To rectify the aforementioned limitations, here we propose a novel soft block pruning approach i.e. *Soft **U**niform **B**lock **P**runing* (SUBP) to obtain high accuracy sub-model without a pre-trained super-model or fine-tuning the pruned model. In contrast to the traditional hard pruning approaches, we recover the pruned blocks through an importance sampling approach, which enables the compressed network to have a large model capacity and thus achieves a higher accuracy than others. As shown in Fig. 2, the details of our SUBP can be divided into two stages in what follows.

**1) Block Pruning Stage:** Firstly, we init all elements in $M_{j,k}^i$ to 1. For the $i$-th layer, we use Eq. (4) to evaluate the importance of each block and identify the set of important blocks to retain as:

$$\mathcal{T}_j^i = \text{ArgTopK}\left(\{S_{j,k}^i | 1 \leq k \leq C_i\}\right) \in \mathbb{R}^{\lceil C_i(1-p_i) \rceil} \tag{5}$$

which gives the indices of blocks with the top $\lceil C_i(1-p_i) \rceil$ scores of $B_j^i$. Then, we prune the bottom-ranking block by zeroizing $\{M_{j,k}^i | k \notin \mathcal{T}_j^i\}$.

**2) Block Regrowing Stage:** To determine the blocks to regrow, we introduce an importance sampling strategy based on the block importance score. We compute the importance of sampling probabilities by $p_{j,k}^i = \exp\left(\frac{S_{j,k}^i}{\tau}\right) / \sum_{m \notin \mathcal{T}_j^i} \exp\left(\frac{S_{j,m}^i}{\tau}\right)$, where the temperature $\tau$ is to balance the sampling attention between the different blocks. Then we select the regrow block indices by performing importance sampling $\mathcal{G}_j^i = \text{Multinomial}(\{p_{j,k}^i | k \notin \mathcal{T}_j^i\}, \delta_t C_i)$ without replacement based on the regrowing factor $\delta_t$ at $t$-th epoch and reset $\{M_{j,k}^i | k \in \mathcal{G}_j^i\}$ to 1.

To compute the regrowing factor $\delta_t$, we employ a gradually schedule [67] to gradually reduce the regrown blocks so that the subnet can converge to the target uniform 1×N sparsity stably at the end of training. Specially, the regrowing factor at $t$-th epoch is computed as:

$$\delta_t = \begin{cases} 1-p, & t \leq t_s \\ \delta_0(1 - \frac{t-t_s}{t_e-t_s})^3, & t_s < t \leq t_e \\ 0, & t_e < t \end{cases} \tag{6}$$

where $\delta_0$ is the initial regrowing factor, $t_s$ and $t_e$ denote the start and end of the training epochs in the block pruning-regrowing stage respectively. Since the model is trained from scratch and the pruning and regrowing decision based on the weights may not be sufficient enough, therefore, we don't prune blocks in the early stage of training. When $t \geq t_e$, the blocks will stop regrowing and we finish pruning the block. The remaining blocks are considered to be the best candidates in the end.

## 4 Experiments

### 4.1 Experiment Settings and Implementation Details

In this section, we evaluate our SUBP on the largescale dataset ImageNet with the representative networks, including ResNet18, ResNet34, ResNet50 and MobileNetV1. We implement SUBP using the PyTorch framework and NVIDIA RTX 3090 GPUs. The SGD optimizer with a momentum of 0.875 and a weight decay of 3e-5 is adopted. We train all compared networks for 250 epochs with a mini-batch size of 512 and an initial learning rate of 0 which is linearly increased to 0.512 during the first 8 epochs and then decayed to 0 by the cosine learning rate schedule. Same as previous methods [18, 45], we also use label smoothing with the factor 0.1 and apply the standard data augmentation. To keep our method simple and generic, the hyper-parameters ($\tau$, $\lambda$, $\delta_0$, $t_s$, $t_e$) in above are set to (1.0, 1.0, 0.2, 10, 180) and kept constant in our experiments. By adjusting appropriate

Table 1: Performance comparison of our BPAR prune criterion against $\ell_1$ norm. The experiment is conducted using MobileNetV1 and ResNet50 with the pruning rate $p = 50\%$ on ImageNet dataset. We test our BPAR on three cases, which N is set to 8, 16 and 32 respectively.

| | MobileNetV1 (p = 50%) | | | | ResNet50 (p = 50%) | | | |
|---|---|---|---|---|---|---|---|---|
| Accuracy | Top-1 | Top-5 | Top-1 | Top-5 | Top-1 | Top-5 | Top-1 | Top-5 |
| Origin | 71.15 | 89.83 | 71.15 | 89.83 | 77.00 | 93.65 | 77.01 | 93.65 |
| Weight Pruning | 70.76 | 89.59 | 70.76 | 89.59 | 77.09 | 93.61 | 77.09 | 93.61 |
| Filter Pruning | 65.35 | 86.26 | 65.35 | 86.26 | 75.38 | 92.52 | 75.38 | 92.52 |
| Prune Criterion | $\ell_1$ norm | | BPAR | | $\ell_1$ norm | | BPAR | |
| Accuracy | Top-1 | Top-5 | Top-1 | Top-5 | Top-1 | Top-5 | Top-1 | Top-5 |
| 1×2 Pattern | 70.28 | 89.37 | - | - | 76.65 | 93.47 | - | - |
| 1×4 Pattern | 70.05 | 89.06 | - | - | 76.51 | 93.24 | - | - |
| 1×8 Pattern | 69.91 | 89.08 | **70.55** | **89.42** | 76.15 | 93.13 | **76.70** | **93.51** |
| 1×16 Pattern | 69.56 | 88.93 | **70.30** | **89.35** | 76.25 | 93.08 | **76.52** | **93.20** |
| 1×32 Pattern | 69.54 | 88.80 | **70.03** | **89.23** | 75.96 | 92.95 | **76.31** | **93.18** |

Table 2: Results of ResNet18, ResNet34, ResNet50 and MobileNetV1 on ImageNet dataset. "PT": require pre-training. "SR": sparse ratio.

| Method | PT | FLOPs | SR | Top-1 | Epochs | Method | PT | FLOPs | SR | Top-1 | Epochs |
|---|---|---|---|---|---|---|---|---|---|---|---|
| ResNet-18 | | | | | | ResNet-50 | | | | | |
| PFP [39] | ✓ | 1.27G | 43.8% | 67.4% | 270 | SSS [31] | ✗ | 2.3G | 38.8% | 71.8% | 100 |
| SCOP [59] | ✓ | 1.10G | 39.3% | 69.2% | 230 | TAS [11] | ✗ | 2.3G | 43.5% | 76.2% | 240 |
| SFP [24] | ✓ | 1.04G | 47.6% | 67.1% | 200 | GAL [43] | ✓ | 2.3G | 16.8% | 72.0% | 150 |
| FPGM [25] | ✓ | 1.04G | 47.6% | 68.4% | 200 | Hrank [40] | ✓ | 2.3G | 36.7% | 75.0% | 570 |
| DMCP [18] | ✗ | 1.04G | 17.6% | 69.0% | 150 | Taylor [51] | ✓ | 2.2G | 44.3% | 74.5% | - |
| CHEX [29] | ✗ | 1.03G | 38.7% | 69.6% | 250 | C-SGD [6] | ✓ | 2.2G | 42.8% | 74.9% | - |
| **SUBP(1×16)** | ✗ | 1.03G | 44.1% | **69.9%** | 250 | SCOP [59] | ✓ | 2.2G | 42.8% | 76.0% | 230 |
| **SUBP(1×32)** | ✗ | 1.03G | 44.1% | **69.7%** | 250 | DSA [52] | ✗ | 2.0G | - | 74.7% | 120 |
| ResNet-34 | | | | | | CafeNet [57] | ✗ | 2.0G | 27.8% | 76.9% | 300 |
| Taylor [51] | ✓ | 2.8G | 21.1% | 72.8% | - | CHEX [29] | ✗ | 2.0G | 35.8% | 77.4% | 250 |
| SFP [24] | ✓ | 2.2G | 49.1% | 71.8% | 200 | **SUBP(1×16)** | ✗ | 2.0G | 45.5% | **77.6%** | 250 |
| FPGM [25] | ✓ | 2.2G | 49.1% | 72.5% | 200 | **SUBP(1×32)** | ✗ | 2.0G | 45.5% | **77.4%** | 250 |
| GFS [62] | ✓ | 2.1G | 32.5% | 72.9% | 240 | SCP [32] | ✗ | 1.9G | - | 75.3% | 200 |
| DMC [15] | ✓ | 2.1G | - | 72.6% | 490 | Hinge [38] | ✓ | 1.9G | - | 74.7% | - |
| NPPM [14] | ✓ | 2.1G | - | 73.0% | 390 | AdaptDCP [69] | ✓ | 1.9G | 51.5% | 75.2% | 210 |
| SCOP [59] | ✓ | 2.0G | 45.6% | 72.6% | 230 | LFPC [23] | ✓ | 1.6G | - | 74.5% | 235 |
| CafeNet [57] | ✗ | 1.8G | 21.1% | 73.1% | 300 | ResRep [8] | ✓ | 1.5G | - | 75.3% | 270 |
| CHEX [29] | ✗ | 2.0G | 29.2% | 73.5% | 250 | Polarize [68] | ✓ | 1.2G | - | 74.2% | 248 |
| **SUBP(1×16)** | ✗ | 2.0G | 43.8% | **73.7%** | 250 | DSNet [36] | ✓ | 1.2G | - | 74.6% | 150 |
| **SUBP(1×32)** | ✗ | 2.0G | 43.8% | **73.6%** | 250 | CURL [47] | ✓ | 1.1G | 73.8% | 73.4% | 190 |
| MobileNetV1 | | | | | | DMCP [18] | ✗ | 1.1G | 43.6% | 74.1% | 150 |
| 0.75x [30] | ✗ | 325M | 38.1% | 68.4% | - | MetaPrune [45] | ✗ | 1.0G | 53.5% | 73.4% | 160 |
| NetAdapt [61] | ✓ | 284M | - | 69.1% | - | EagleEye [35] | ✓ | 1.0G | 69.4% | 74.2% | 240 |
| AMC [26] | ✓ | 285M | - | 70.5% | - | CafeNet [57] | ✗ | 1.0G | 52.9% | 75.3% | 300 |
| MetaPruning [45] | ✓ | 281M | 50.9% | 70.6% | 320 | CHEX [29] | ✗ | 1.0G | 67.1% | 76.0% | 250 |
| **SUBP(1×16)** | ✗ | 279M | 40.0% | **70.8%** | 250 | **SUBP(1×16)** | ✗ | 1.0G | 68.3% | **76.3%** | 250 |
| **SUBP(1×32)** | ✗ | 279M | 40.0% | **71.1%** | 250 | **SUBP(1×32)** | ✗ | 1.0G | 68.3% | **76.0%** | 250 |

parameters for different models and pruning rates, better results should be obtained in general. In this paper, we calculate FLOPs by counting multiplication and addition as one operation as He [22] did.

## 4.2 Influence of Pruning Criterion

In this subsection, we will investigate the influence of 1×N pruning criterion. Table 1 shows the compared results with respect to $\ell_1$ norm [42] and BPAR. As suggested by [37, 48, 51, 28], the pruning criterion selects the appropriate sub-model on the basis of its importance or redundancy, which plays an important role in the traditional PPF approach. Table 1 tells us that our BPAR

outperforms the norm-based method on the ImageNet dataset. For MobileNetV1 and ResNet50, our BPAR can achieve consistent accuracy improvements with the same sparsity and inference speedup. On MobileNetV1(1×16), it obtains 0.74% and 0.38% top1-accuracy and top5-accuracy improvement when BPAR is applied. In particular, for pruning a pre-trained MobileNetV1, BPAR achieves better results for N=8 than $\ell_1$ norm for N=2, which indicates 1×N pruning based on BPAR can obtain a better trade-off between accuracy and inference latencies. Similar results can also be observed on the ResNet50 in Table 1. In essence, our BPAR explicitly utilizes the relationship and identifies the mutual angular redundancy between blocks, giving rise to its superior performance.

### 4.3 Results on ImageNet

To verify the effectiveness of our SUBP, we apply it to the heavyweight CNN model (i.e. ResNet [22]) with different depths and the lightweight CNN model (i.e. MobileNet) initialized with random weights on ImageNet [5] dataset. Here, ResNet-18/34/50 and MobileNetV1 are used as the baseline models and have 1.8/3.7/4.1 GFLOPs and 572 MFLOPs.

The results in Table 2 show that SUBP achieves noticeably higher accuracy than the state-of-the-art pruning methods under the same FLOPs constraints. For example, our SUBP on ResNet50(1×16) with 2× FLOPs reduction achieves 77.6% top-1 accuracy, which is 3.1%, 1.6%, 0.7% and 0.2% higher than Taylor[51], SCOP [59], CafeNet [57] and CHEX [29] respectively. The results on 1×32 sparsity also demonstrate the superiority against the other methods. On the other hand, at the same target accuracy, our SUBP also achieves higher FLOPs reduction. For example, our SUBP achieves 4× FLOPs reduction and 76.0% top-1 accuracy on ResNet50(1×32) compared to the SCOP, which only yields 1.9× FLOPs reduction. We further observe that SUBP also achieves higher accuracy at a fraction of the training cost for MobileNetV1. For instance, SUBP achieves 71.1% top-1 accuracy on MobileNetV1(1×32, 250 epoch) under the 279M FLOPs constraint, which is 0.5% higher than MetaPruing [45](320 epoch). This is because SUBP can dynamically explore the optimal sub-model in one training pass from scratch, circumventing the expensive PPF cycles.

### 4.4 Results on Object Detection and Instance Segmentation

To further explore the performance of SUBP in downstream tasks, we conduct experiments on object detection and instance segmentation on the challenging COCO dataset [44]. The results are shown in Table 3 and Table 4. We adopt the classical method Faster RCNN [54] for object detection and Mask RCNN [21] for instance segmentation. We use ResNet50 with different sparse ratios and block sizes as backbone. All the experiments are conducted based on MMDetection [1]. Compared to Dense ResNet50, the SUBP can achieve competitive results, which further demonstrates itsrobustness and superiority on downstream computer vision tasks.

Table 3: Object detection results on COCO.

| Model | Block Size | mAP |
|---|---|---|
| F-RCNN-R50(4.1G) | - | 37.4 |
| F-RCNN-R50(2.0G) | 1× 32 | 38.4 |
| F-RCNN-R50(2.0G) | 1× 16 | 38.5 |
| F-RCNN-R50(1.0G) | 1× 32 | 37.1 |
| F-RCNN-R50(1.0G) | 1× 16 | 37.3 |

Table 4: Instance segmentation results on COCO.

| Model | Block Size | Box mAP | Mask mAP |
|---|---|---|---|
| M-RCNN-R50(4.1G) | - | 38.2 | 34.7 |
| M-RCNN-R50(2.0G) | 1× 32 | 39.2 | 35.4 |
| M-RCNN-R50(2.0G) | 1× 16 | 39.4 | 35.5 |
| M-RCNN-R50(1.0G) | 1× 32 | 37.4 | 33.8 |
| M-RCNN-R50(1.0G) | 1× 16 | 37.5 | 33.8 |

### 4.5 SUBP from Pre-trained Models

In order to further investigate the generality property of our approach, we apply SUBP to the model initialized with pre-trained weights. For an equitable comparison with other PPF methods, we adopt the pre-trained ResNet18 provided by the torchvision[2] and run SUBP for 120 epochs as the previous one did. The results in Table 5 show that our SUBP still achieves competitive top-1 and top-5 accuracy under the same FLOPs compared to the previous state-of-the-art PPF methods. Furthermore, when ResNet18 is trained for 90+120 epochs, it achieves 69.5% top-1 accuracy, which is only 0.4% lower than training it from scratch for 250 epochs in Table 2.

---

[2]https://pytorch.org/vision/stable/models.html

Table 5: ResNet18 starting from the pre-trained models on ImageNet dataset. "Epochs" are reported as: pre-training epochs plus all subsequent training epochs needed to obtain the final pruned model.

| Model | Method | Params | FLOPs | Top-1 | Top-5 | Epochs |
|-------|--------|--------|-------|-------|-------|--------|
| ResNet-18 | Baseline | 11.7M | 1.81G | 69.8% | 89.1% | 90 |
| | PFP [39] | 6.6M | 1.27G | 67.4% | 87.9% | 90+180 |
| | SCOP [59] | 7.1M | 1.10G | 69.2% | 88.9% | 90+140 |
| | SFP [24] | 7.1M | 1.04G | 67.1% | 87.8% | 100+100 |
| | FPGM [25] | 7.1M | 1.04G | 68.4% | 88.5% | 100+100 |
| | CHEX [29] | - | 1.04G | 69.2% | - | 90+120 |
| | SUBP(1×16) | 7.1M | 1.04G | **69.5%** | **89.0%** | 90+120 |
| | SUBP(1×32) | 7.1M | 1.04G | **69.3%** | **88.9%** | 90+120 |

## 4.6 Deployment Efficiency Analysis

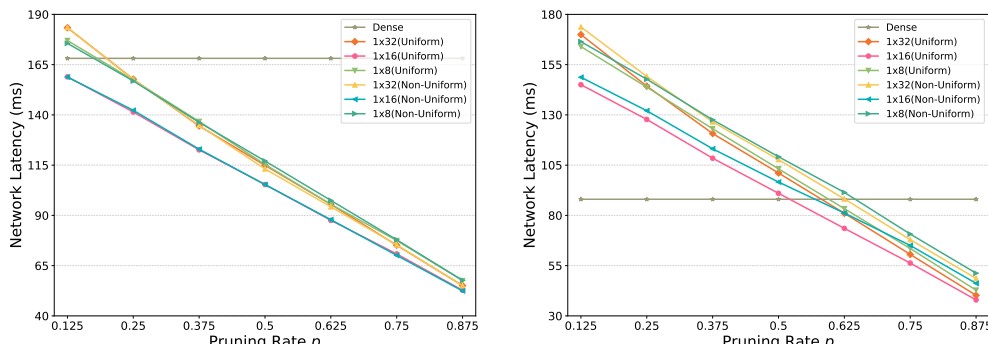

Figure 4: Network latency comparison between uniform 1×N sparse against non-uniform and dense model with varying N and prune rates. The experiment is conducted using ResNet18 and set the input shape as (4, 3, 224, 224) on the arm platform of Apple M1 Pro CPU @ 3.20GHz with single thread (left) and two threads (right). Best viewd in colors.

To further explore the acceleration capacity with respect to different N and prune rates on CPUs-based platforms, we adopt TVM [2] to compile the uniform and non-uniform 1×N pruned models. For a fair comparison, we also consider TVM for dense model to obtain baseline latency. We deploy the 1×N pruned models to obtain network latencies on the arm platform of Apple M1 Pro CPU @ 3.20GHz. From Fig. 4, we observe 1×N pruned model achieves noticeable latency reductions across various pruning rates and N. For example, when the pruning rate is greater than 0.25, the inference latencies of different N are all the better than their dense baseline under the single thread scenario. When the thread num is set to 2, the inference speed of different configurations has been improved. Due to the workload imbalance between threads in non-uniform 1×N, it always lags behind its uniform counterpart. From Fig. 4, there are also two points worth noting: 1) the improvement of 1×N is not as significant as a vanilla convolution in a multithreading context; 2) N=16 achieves a faster inference speed than other N on M1 Pro. Therefore, optimizing multithreading inference with 1×N pruning and selecting appropriate N based on suitable platforms are directions that can be further explored.

## 5 Limitation and Discussion

Firstly, our SUBP masks weights throughout the training and still needs dense computations. Therefore, while SUBP has advantages over methods that depend on PPF pipeline, there is still room for improving its training efficiency. Secondly, there are still missing experiments, including applying 1×N sparsity on other types of DNNs like RNN, transformer and other tasks including object detection, natural language processing, *etc.*. We will design corresponding high-performance operators to improve the training efficiency of SUBP, and explore the performance of 1×N sparsity in other types of DNNs and tasks to verify its broader applicability in our future work.

# 6  Conclusion

Uniform $1 \times N$ sparsity is an important technique that allows fast inference on multithreading scenarios under multi-core architecture. This paper proposes soft uniform block pruning, SUBP, to efficiently train a uniform $1 \times N$ sparsity network from scratch. SUBP dynamically adjusts the pruning blocks based on a periodic pruning and regrowing process, which prevents the important blocks from being prematurely pruned and keeps the model's capacity. We also present a new block pruning criterion named BPAR, which explicitly considers the mutual angular redundancy between blocks rather than "relatively less" importance only. By proposing a BPAR based block subset selection approach for pruning and an importance sampling based strategy for regrowing, we can obtain a sub-model with high accuracy by end-to-end implementation without pre-training a large model or requiring extra fine-tuning. Extensive experiments have exhibited our method can effectively reduce the FLOPs and inference latencies while achieving superior accuracy over several SOTAs.

## Acknowledgments and Disclosure of Funding

This work is funded in part by the Key Research and Development Project of Zhejiang Province under Grant 2021C01035.

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

## A  Supplementary Material

This **Supplementary Material** is structured as follows. We provide a formulation of our algorithm in Section B. To investigate the effectiveness of different components of our SUBP, we conduct ablation studies and provide additional experimental results in Section C. In Section D, we provide deployment results on the x86 platform of Intel(R) Xeon(R) Platinum 8260L CPU @ 2.30GHz to further explore the performance of $1\times$N sparse on different platforms. Finally, in Section E, we discuss the societal impact of our method.

## B  Algorithm Formulation

---

**Algorithm 1:** Overview of the SUBP method.

---

1 **Input**: An $L$-layer CNN model with weights $\mathbf{W} = \{W^i | 1 \leq i \leq L\}$; block binary mask matrices $\mathbf{M} = \{M^i_{j,k} \in \{0,1\} | 1 \leq i \leq L, 0 \leq j \leq \frac{C_{i+1}}{N} - 1, 0 \leq k \leq C_i - 1\}$; indices of activate blocks with the top scores $\mathbf{T}$; indices of regrow blocks based on importance sampling $\mathbf{G}$; target prune rate $p$; initial regrowing factor $\delta_0$; importance balance coefficient $\lambda$; sampling attention balance factor $\tau$; training epochs $T_{\text{total}}$; start and end epoch in the pruning-regrowing stage $t_s, t_e$; training set $\mathcal{D}$ ;

2 **Output**: A sub-model satisfying the target prune rate $p$, its optimal weight values $\mathbf{W}^*$ and binary mask $\mathbf{M}^*$ ;

3 Randomly initialize the model weights $\mathbf{W}$;

4 Initialize $\{M^i_{j,k} \mid \forall i, \forall j, \forall k\}$ to 1 ;

5 Reformat $\mathbf{W}$ to $\mathbf{B}$ according to Section 3 ;

6 **for** *each training epoch $t \in [T_{total}]$* **do**

7      Sample a mini-batch from $\mathcal{D}$ and update the model weights $\mathbf{W}$ ;

8      **if** $t_s < t \leq t_e$ **then**

9          Reset $\{M^i_{j,k} \mid \forall i, \forall j, \forall k\}$ to 1 ;

10          Compute the importance score $S$ of block by Eq. 4 ;

11          Get the indices of activate blocks with the top scores by Eq. 5 ;

12          Prune the bottom-ranking block by set $\{M^i_{j,k} | k \notin \mathcal{T}^i_j\}$ to 0;

13          Compute the importance sampling probabilities by
$$p^i_{j,k} = \exp\left(\frac{S^i_{j,k}}{\tau}\right) / \sum_{m \notin \mathcal{T}^i_j} \exp\left(\frac{S^i_{j,m}}{\tau}\right) ;$$

14          Compute the regrowing factor by Eq. 6 ;

15          Get the indices of regrow blocks based on importance sampling by
$$\mathcal{G}^i_j = \text{Multinomial}(\{p^i_{j,k} | k \notin \mathcal{T}^i_j\}, \delta_t C_i) \text{ without replacement} ;$$

16          Regrow the blocks by resetting $\{M^i_{j,k} | k \in \mathcal{G}^i_j\}$ to 1 ;

---

## C  Ablation Analysis

Table 6: Compare different design choices in the regrowing stages of the SUBP method. All the experiments are based on the TinyImageNet with ResNet18($1\times32$). The random baseline is 57.0%.

| *Regrowing factor* | | | | |
|---|---|---|---|---|
| Design choices | $\delta_0 = 0.1$ | $\delta_0 = 0.2$ | $\delta_0 = 0.3$ | $\delta_0 = 0.4$    Full |
| Top-1 | 57.6% | 58.4% | 57.9% | 58.0%    58.5% |

| *Decay scheduler for block regrowing* | | | |
|---|---|---|---|
| Design choices | Default | Constant | Linear decay    Cosine decay |
| Top-1 | 58.3% | 57.5% | 58.4%      58.3% |

In Table 6, we investigate the effectiveness of different design choices in our block regrowing stage. All the experiments are based on the TinyImageNet with ResNet18(1×32). Compared to the random

baseline with 57.0% top-1 accuracy, our SUBP achieves consistent improvement under the different settings.

We find that regrowing factor $\delta_0$ significantly impacts the final quality of the model. Intuitively, a larger regrowing factor can provide a more extensive sampling space during training and retain the model's capacity to a greater extent. However, a sizeable regrowing factor may also cause drastic sub-model structure changes, affecting stability during training. As shown, the accuracy is improved by 0.8% as the $\delta_0$ increases from 0.1 to 0.2. When $\delta_0$ increases again, the model's accuracy drops until $\delta_0$ is the full model size. This suggests that the relationship between the regrowing factor and the final quality of the model is varied, and selecting an appropriate regrowing factor in specific circumstances can improve the final quality.

We also investigate the decay scheduler for the block regrowing stage. We compare several decay schedulers, including default (Eq. 6), constant, linear, and cosine. The experiments show SUBP has good robustness to different decay schedulers, as default, linear, and cosine decay schedulers all show similar performance. With a decay scheduler, the sampling space can be gradually decreased, and the sub-model under training can converge stably.

Table 7: Ablation study of block size and epochs. All the experiments are based on the ImageNet with ResNet18.

| Method | FLOPs | Accuracy | Epochs |
|---|---|---|---|
| DMCP | 1.04G | 69.0% | 150 |
| CHEX | 1.03G | 69.6% | 250 |
| SUBP ($1\times4$) | 1.03G | 70.4% | 250 |
| SUBP ($1\times8$) | 1.03G | 70.2% | 250 |
| SUBP ($1\times16$) | 1.03G | 68.7% | 100 |
| SUBP ($1\times16$) | 1.03G | 69.2% | 150 |

To achieve a fairer and fuller comparison with previous methods, we conduct ablation study with the same epoch and different block sizes. It can be seen in Table 7 that both longger training epochs and finer pruning granularity can improve the final results. We can weight trade offs in practical applications.

## D  Deployment on x86 Platform

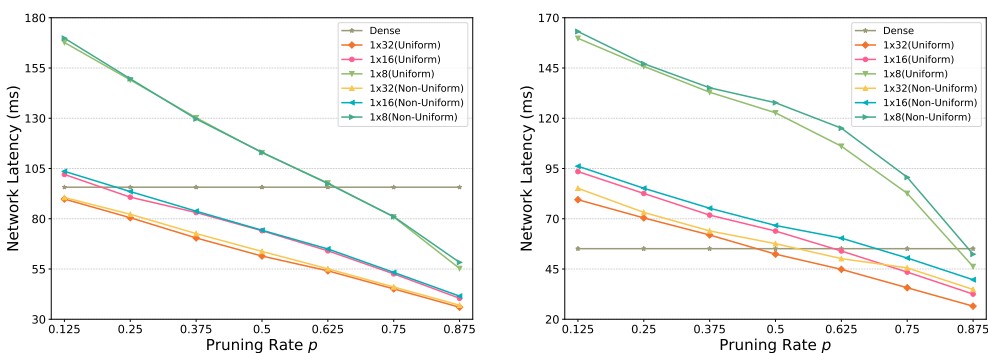

Figure 5: Network latency comparison between uniform $1\times N$ sparse against non-uniform and dense model with varying N and prune rates. The experiment is conducted using ResNet18 and set the input shape as (4, 3, 224, 224) on the x86 platform of Intel(R) Xeon(R) Platinum 8260L CPU @ 2.30GHz with single thread (left) and two threads (right). Best viewd in colors.

In order to further explore the performance of $1\times N$ sparse DNNs on different platforms, as shown in Fig. 5, we also conducted corresponding experiments on the x86 platform of Intel(R) Xeon(R) Platinum 8260L CPU @ 2.30GHz and obtain the similar results in general: 1) The gain of vanilla convolution in multithreading scenarios is much greater than that of $1\times N$ sparse convolution. 2) The

inference speed of uniform $1\times N$ is slightly faster than that of non-uniform in the case of multithreading, indicating the importance of workload balance again. However, unlike the performance on the arm platform of Apple M1 Pro CPU @ 3.20GHz, the $1\times N$ sparse DNNs are significantly accelerated when N is set to 16 and 32 on the Platinum 8260L CPU @ 2.30GHz. We can also notice that in most cases, N=32 achieves a fast inference speed.

# E  Societal Impact

Our method can reduce the computational overhead of training and inferencing stages while achieving satisfactory accuracy on modern CNN models. This can facilitate the application of CNN models on edge devices and is of high value for the community and society to realize Green AI. At the same time, our $1\times N$ sparse DNNs are based on new sparse operators, which can promote the progress of related hardware and algorithms to a certain extent.

