# OpenReview forum: "SUBP: Soft Uniform Block Pruning for 1$\times$N Sparse CNNs Multithreading Acceleration"
_NeurIPS.cc/2023/Conference — NeurIPS 2023 poster_

### Official Review · Reviewer_Ah3e · 2023-06-27

**Soundness:** 3 good
**Presentation:** 3 good
**Contribution:** 3 good
**Rating:** 6
**Confidence:** 5

**Summary:**

This paper proposes a new pruning approach to train a uniform 1×N sparse structured network from scratch, dubbed as SUBP. SUBP pruned blocks according to block angular redundancy. The blocks are pruned in a uniform manner to make full use of multithread parallelism computation. During the training process, the pruned blocks can be regrown through importance sampling. They conduct experiments across various CNN architectures to show the superiority of  SUBP.

**Strengths:**

- This paper is well-written and its motivation is reasonable.
- Code seems to be available, and it'll be helpful if the authors could open-source the work if the paper is accepted.
- They deploy the pruned model in both the arm platform and x86 platform to get actual inference speed. These results well-validate that uniform pruning methods are faster than non-uniform pruning methods.

**Weaknesses:**

-  The authors are encouraged to conduct experiments on other tasks, not just classification.
-  The authors only report the FLOPs in Table 2, the authors are encouraged to report the training time.
-  Compared with method CHEX, the improvements in accuracy are limited.

**Questions:**

- Please see the weaknesses part.
- Why the inference latency of the sparse model is lower than the inference latency of the dense model in low pruning rates?

**Limitations:**

As discussed in Section 5, there are still missing experiments, including applying  1×N sparsity on other types of DNNs like RNN, transformer, and other tasks including object detection, natural language processing, etc.. The generalization on downstream tasks still needs to be verified.

---

> ### Author Rebuttal · Authors · 2023-08-06
>
> Thanks for your constructive and supportive comments.
>
> **Q1:** The authors are encouraged to conduct experiments on other tasks.
>
> **A1:** Following your advice, we further conduct experiments on two other tasks including object detection and instance segmentation. Below displays the experimental results in which the proposed SUBP performs well on various vision tasks.
>
> **Object detection results on COCO:**
>
> |**Model**|**Block Size**|**mAP**|
> |-|-|-|
> |F-RCNN-R50 (4.1G)|-|37.4|
> |F-RCNN-R50 (2G)|1x32|38.4|
> |F-RCNN-R50 (2G)|1x16|38.5|
> |F-RCNN-R50 (1G)|1x32|37.1|
> |F-RCNN-R50 (1G)|1x16|37.3|
>
> **Instance segmentation results on COCO:**
>
> |**Model**|**Block Size**|**Box mAP**|**Mask mAP**|
> |-|-|-|-|
> |F-RCNN-R50 (4.1G)|-|38.2|34.7|
> |F-RCNN-R50 (2G)|1x32|39.2|35.4|
> |F-RCNN-R50 (2G)|1x16|39.4|35.5|
> |F-RCNN-R50 (1G)|1x32|37.4|33.8|
> |F-RCNN-R50 (1G)|1x16|37.5|33.8|
>
> **Q2**: The authors only report the training FLOPs, the authors are encouraged to report the training time.
>
> **A2**: The training time for sparsifying ResNet-18/34/50 on ImageNet is provided in the following.
>
> |Model|Block size|Training Time (NVIDIA RTX 3090 GPU days)|
> |-|-|-|
> |ResNet18|1x16|1.84|
> |ResNet18|1x32|1.83|
> |ResNet34|1x16|2.88|
> |ResNet34|1x32|2.87|
> |ResNet50|1x16|3.61|
> |ResNet50|1x32|3.59|
>
> **Q3:** The authors' method improves accuracy to a limited extent compared to CHEX.
>
> **A3:** Your question is very valuable and we have realised this point in the course of our experiments. We think this reason may come from **two points:** 1. CHEX applies different sparsity ratios to different layers, using the FLOPs of the whole network as pruning targets. Different layers of a neural network have different redundancies, so applying different sparsity ratios to different layers tends to yield better results. Our approach applies the same sparse ratio to each layer of the network; 2. We apply coarser-grained block pruning (1x32, 1x16) to 1xN, and we conducted 1x8 and 1x4 experiments on ResNet18, which yielded 70.2% and 70.4% top-1 accuracy on ImageNet, respectively, compared to CHEX's 69.6%, and, as can be seen in Fig. 4, the 1x8 pruning granularity also achieves better inference latency on hardware.
>
> **Q4:** Why the inference latency of the sparse model is lower than the inference latency of the dense model in low pruning rates.
>
> **A4:** The pattern of sparse computation is very different from that of dense computation. For dense matrix multiplication, GEMM is currently the most dominant method. Sparse matrix multiplication can not make full use of cache like GEMM, so its computational efficiency is very much related to the sparse algorithm and the pattern of sparse matrix, and researchers have carried out a lot of studies, such as spmm, spmv and so on. Current sparse matrix multiplication methods have a sparsity threshold for speedup, above which the computational delay of sparse matrices outperforms that of dense matrix multiplication only. The more efficient sparse matrix multiplication corresponds to a lower sparsity threshold.

---

> > ### Comment · Reviewer_Ah3e · 2023-08-19
> >
> > Thank you for the author's response. Your answer has addressed my concerns, and I will maintain a positive rating. I am looking forward to a revised version of the paper with the experiments on downstream tasks.

---

### Official Review · Reviewer_nWst · 2023-07-06

**Soundness:** 2 fair
**Presentation:** 1 poor
**Contribution:** 1 poor
**Rating:** 3
**Confidence:** 5

**Summary:**

SUBP trains a uniform 1×N sparse CNNs from scratch, with three characteristics: (1) a periodic block pruning and regrowing technique via importance sampling, (2) a pruning criterion based on angular redundancy across blocks,  and (3) a uniform 1×N sparse pattern for multithreading acceleration. The authors also provide latency results on hardware platforms.

**Strengths:**

1. The method achieves competitive performance against many existing methods.

2. SUBP trains a uniform 1×N sparse CNNs from scratch.

**Weaknesses:**

1. Novelty concern. This paper uses a prune-and-grow idea to train a sparse model from scratch, which is been well-studied by many papers in sparse training [1-2]. DMCP and CHEX also don’t require pre-training. Work [3] studied how to prune the network from scratch with a grow-and-prune methodology. Besides, the 1xN pruning technique needs compiler optimization (like TVM) to achieve real inference acceleration. Therefore, the acceleration part of this work highly relies on TVM.

2. The performance improvements are marginal. In Table 1, it’s hard to tell if SUBP is better than CHEX. More important, CHEX uses channel pruning, which could achieve a faster acceleration rate when the pruning ratio is the same. From my point of view, CHEX is basically faster and has the same performance in terms of accuracy. DMCP (channel pruning) is only trained 150 epochs, which is also hard to tell if SUBP has better performance (69.0%,150 epochs vs 69.7%, 250 epochs). Overall, when compared with many SOTA methods, SUBP could not achieve a faster acceleration rate or has an obviously better performance in terms of accuracy. Table 2 has the same problem with many outdated baselines. The authors should provide error bars when the accuracy improvements are marginal.

3. The paper is written poorly. Sections 1 and 3 should be organized better, and section 3 is not clearly written. Table 1 first appeared on Page 6 (section 3) and was first mentioned at the end of Page 7 (section 4). Table 2 has an error with vertical lines exceeding the table. Also, the font size of the figures in this paper is very small, especially in Figure 2. All these things make this paper hard to read.

4. There is no ablation study to show the effectiveness of each components of SUBP.

[1] Rigging the Lottery: Making All Tickets Winners, ICML 2020.

[2] MEST: Accurate and Fast Memory-Economic Sparse Training Framework on the Edge, Neurips 2021.

[3] Effective Model Sparsification by Scheduled Grow-and-Prune Methods, ICLR 2022.

**Questions:**

Please refer to “Weaknesses".

**Limitations:**

Please refer to “Weaknesses" for limitations. The authors adequately addressed the potential negative societal impact of their work.

---

> ### Author Rebuttal · Authors · 2023-08-06
>
> Thanks for your in-depth review that will help us strengthen the manuscript. We hope our response can address your concern here.
>
> **Q1:** Prune-and-regrow idea has been well-studied and acceleration of TVM is highly relies on TVM.
>
> **A1:** With all due respect, though the concept of grow-and-prune exists in other methods, we would like to highlight that our SUBP is different from the previous method on several aspects. **In terms of pruning strategy**, our approach uses importance sampling based approach for the weights that have been removed, unlike the deterministic strategy of selecting topk importance weights for resurrection in [1-3]; **in terms of weight/channel importance measure**, [1-4] use the magnitude of the weights, norm-based criterion or matrix decomposition-based importance criterion, our method takes into account the angular redundancy of the weight blocks, which is different from the previous methods; **in terms of granularity of pruning**, we study 1xN block pruning, while CHEX [4], DMCP [5] focussing on channel pruning and [1] focussing on weight pruning.
>
> **For the acceleration of 1xN block pruning,** on CPU devices, in addition to TVM[6], MNN[7] and XNNPACK[8] provide corresponding acceleration operators. On GPU devices, CVW[9] has also experimentally demonstrated that 1xN block pruning on GPUs has equally great potential.
>
> **Q2:** The authors' method does not improve significantly enough compared to CHEX and is not as efficient as CHEX or other SOTA methods with channel pruning. Also, compared to DMCP, the authors' method trains more epochs, making it difficult to state whether the method has better performance.
>
> **A2:** **In response to the issue that the accuracy of our method compared CHEX is marginal,** we believe there are two reasons: **firstly**, since different layers of the neural network have different degrees of redundancy, methods like CHEX and DMCP that apply different sparsity ratios to different layers tend to achieve higher results; **secondly**, since our methods all apply a coarser-grained block structure to the model, when we apply our method When we apply our method to 1x4, 1x8 block pruning, our method achieves better results, while referring to the inference delay in Fig.4, the appropriate fine-grainedness can also achieve efficient inference latency.
>
> **For inference latency**, papers [8-10] have shown that this semi-structured sparse pruning model achieves a better trade-off between accuracy and inference latency compared to channel pruning. It is also important to note that the current channel pruning approach, although it can achieve better efficiency on GPUs, can not be deployed on edge devices like CHEX with single channel pruning because the computational chips on edge devices and deploy framework often require channels to be aligned in multiples of 4 or 8, such as Ascend310 and Bolt, complementary channels incurs additional overheads. Our approach can be deployed on general-purpose hardware with specific sparse operators, and channel patching is not required. In addition, there has been a large amount of research focused on how to improve the efficiency of sparse computing. We believe that in the future, this semi-structured sparse model can achieve better trade-off.
>
> **Regarding the comparison with DMCP**, we train 100 and 150 epochs on ResNet18 (1x16), and we can see that our method can also achieve better performance with similar epochs.
> All the results are shown in the following table.
>
> | Method | FLOPs | Accuracy | Epochs |
> | -| -- | - | - |
> | DMCP | 1.04G | 69.0% | 150 |
> | CHEX | 1.03G | 69.6% | 250 |
> | SUBP (1x4)  | 1.03G | 70.4% | 250 |
> | SUBP (1x8)  | 1.03G | 70.2% | 250 |
> | SUBP (1x16) | 1.03G | 68.7% | 100 |
> | SUBP (1x16) | 1.03G | 69.2% | 150 |
>
> **Q3:** The authors should have organised the chapters more clearly and revised the problematic areas in the figures and tables.
>
> **A3:** First of all, I am very sorry for the problems in the fonts and tables in the figures in this paper that cause you problems in reading, if the paper is accepted we will revise the problems that exist.
>
> **Section 1** focuses on the organization from introducing the background, to the granularity of pruning and the problems that exist at the hardware and deployment level. All of them lead to the soundness of our study of 1xN pruning in this paper. Then we point out the shortcomings of the previous studies, and finally present our approach.
>
> **In Section 3** we organize along the definition of 1xN block pruning, block pruning based on angular redundant and prune-and-grow method. We place the relevant algorithm the appendix for better readability.
> Finally, we hope that our explanations will solve your doubts.
>
> **Q4:** Authors should add ablation experiments to demonstrate the effectiveness of each component.
>
> **A4:** We conducted experiments in Table.1 to validate the effectiveness of our pruning criterion compared to norm-based. Also, combining the data in Tables 1 and Tables 2, we verified the effectiveness of the different parts of the methodology. In the Appendix section C, we also added the  ablation experiments of hyper-parameter.
>
> [1] Rigging the Lottery: Making All Tickets Winners, ICML 2020.
>
> [2] MEST: Accurate and Fast Memory-Economic Sparse Training Framework on the Edge, Neurips 2021.
>
> [3] Effective Model Sparsification by Scheduled Grow-and-Prune Methods, ICLR 2022.
>
> [4] CHEX: Channel exploration for CNN model compression. CVPR 2022.
>
> [5] DMCP: Differentiable markov channel pruning for neural networks. CVPR 2020.
>
> [6] TVM: An Automated End-to-End Optimizing Compiler for Deep Learning. OSDI 2018.
>
> [7] MNN: A universal and efficient inference engine. MLSYS 2020.
>
> [8] Fast sparse convnets. CVPR. 2020.
>
> [9] Accelerating Sparse Convolution with Column Vector-Wise Sparsity. NeurIPS 2022.
>
> [10] 1xN Pattern for Pruning Convolutional Neural Networks. TPAMI, 2022.

---

> > ### Comment · Reviewer_nWst · 2023-08-20
> >
> > **I thank the authors' rebuttal. After reading the rebuttal and other reviewers' comments, I will maintain my current score.**
> >
> > First of all, the authors' response does not address my novelty concern, the weight importance measurement techniques were widely used in many pruning papers [1][2], and a heuristic new form does not make it novel. Work [3] also proposes a method to measure angular differences. Second, as the novelty is limited, performance improvement is especially important to demonstrate its effectiveness. **However, the authors' response proves that my previous concern was reasonable**, the accuracy performance improvement is marginal when compared with DMCP and CHEX while using DMCP will benefit more from acceleration during deployment. Furthermore, as the performance improvement is marginal, showing error bars with baseline methods is a typical move.
> >
> > **I disagree with A2 about inference latency.** For practical deployment of pruning, people often tend to use channel pruning and leverage network slimming techniques [4][5]. Since acceleration using semi-structured pruning is often required extra compiler-level techniques (e.g. Nvidia 2:4, 4:8 pruning pattern) to achieve that. However, there is no general backend support for these compiler-level techniques (which means developers need to support them case by case) and the acceleration performance could vary a lot on different devices.
> >
> > **Therefore, given the current state of the paper, I maintain my decision to reject it.**
> >
> > [1] Layer-adaptive sparsity for the magnitude-based pruning, ICLR 2021.
> >
> > [2] MEST: Accurate and Fast Memory-Economic Sparse Training Framework on the Edge, Neurips 2021.
> >
> > [3] A pruning method based on the dissimilarity of angle among channels and filters, arxiv 2022
> >
> > [4] Slimmable Neural Networks, ICLR 2019.
> >
> > [5] Universally Slimmable Networks and Improved Training Techniques, ICCV 2019.

---

> > > ### Author Response · Authors · 2023-08-20
> > >
> > > Thank you very much for your serious reply, here we give further explanations to your concerns about several issues.
> > >
> > > **Q1:** The increase in accuracy compared to DCMP and CHEX is marginal.
> > >
> > > **A1**:  **Firstly**, it is true that increasing the epoch improves the final performance of the model, which has led to unfair comparisons of most pruning approaches.
> > >
> > > **Secondly**, in response to the problem of very small improvement in accuracy, we have explained through experiments in the first rebuttal that it is due to the large block_size we chose, by setting the block_size to 4 and 8, which is more fine-grained, our method achieved significant improvement compared to CHEX in the case of 250 epochs of training, and the specific results can be seen in the first rebuttal.
> > >
> > > **Thirdly**, our approach uses the same pruning rate for each layer of the network, while CHEX and DMCP apply different pruning rates for each layer, which is just intuitively more sensible, since each layer of the network should have a different level of redundancy.
> > >
> > > **In summary**, in addition to the hyperparameters such as the training epoch, the setting of different pruning rates and the design of the pruning block size will affect the performance of the final model. It can be seen from the experimental results that our method has already achieved a significant improvement in accuracy at block sizes of 4 and 8 with the same prune rate among layers. Applying different pruning rates to different layers is vertical to our work. Using different pruning rates for different layers should intuitively yield better results. This is also our future work.
> > >
> > > **Q2**: Concerning about novelty: the components in the author's paper have already appeared in past papers.
> > >
> > > **A2**:  **First of all**, I am very grateful to the reviewers for giving a relevant paper on angular perspective redundancy judgements. We find from our reading that our starting point is similar, but there are still differences: [1] used $cos^{-1}(x1, x2)$ for the redundancy measure while we use $cos(x1,x2)$ after normalization; [1] focuses on filter pruning and only does some experiments on small-scale dataset like cifar100, cifar10 and we focus on 1$\times$N and do experiments on large-scale dataset ImageNet. In the final revised version, we will cite [1] to illustrate the difference between the two specific realizations.
> > >
> > > **Also, regarding novelty**, we acknowledge that the pruning components of our paper have appeared in past papers both in terms of the pruning process and the way importance is measured, but we are combining these components in a reasonable way and applying them to a new problem, where good results have been achieved. From this point of view, we believe that the novelty of our paper is justified.
> > >
> > > **Q3**: On the latency of model deployment.
> > >
> > > **A3**: As reviewer says, channel pruning can go for better performance performance on current general purpose hardware. **However, intuitively, semi-structured pruning should have a higher upper bound on the trade-off between accuracy and performance than structured channel pruning.** And this is the reason why a large number of researchers are currently focusing on semi-structured sparse work.
> > > Admittedly, sparsification requires hardware-level support (e.g. TensorCore) or compiler-level techniques (e.g. TVM), which requires developer support and acceleration performance could vary a lot across  various devices. However, it is because of the existence of so many difficulties that researchers need to carry out corresponding research from the hardware and compiler levels of sparse computing devices, as well as designing efficient algorithms to improve the performance of the model, and make efforts in multiple aspects together to promote the overall development of sparsification.
> > > **In summary, we believe that our semi-structured sparse pruning algorithm has its applications that can lead to better accuracy and performance trade-offs for deployed models in the future as the software and hardware ecosystems develops.**
> > >
> > > Thanks again for your reply, hopefully our response will address your concerns.
> > >
> > > [1] A pruning method based on the dissimilarity of angle among channels and filters, arxiv 2022

---

### Official Review · Reviewer_cAVo · 2023-07-07

**Soundness:** 3 good
**Presentation:** 3 good
**Contribution:** 3 good
**Rating:** 5
**Confidence:** 5

**Summary:**

This paper proposes a new fine-grained structured pruning method, which can train from scratch to reduce the training cost and prunes the weights in a uniform manner to handle the load imbalance problem. During pruning, the method uses block angular redundancy as the criterion to discard the redundant blocks and allows pruned blocks to regrow to the network based on the importance sampling to search for a better sparse pattern. Experiments show that the SUBP method can achieve a significant accuracy improvement compared with other methods under the same FLOPs constraint.

**Strengths:**

- The paper is well-written.
- The proposed SUBP method can greatly improve the model accuracy than kernel pruning methods under a similar pruning ratio.
- The authors propose uniform pruning to handle the load imbalance problem and show the real speedup on the CPU to evaluate the effectiveness of this method.


**Weaknesses:**

- Experiments are insufficient. The authors only evaluate the proposed method on the classification task. It is better to discuss the generalization of the method and evaluate on different tasks.
- The paper lacks a comparison of accuracy with the non-uniform block pruning method. Figure 4 shows that the speedup of the uniform 1xN method is limited. If there is an accuracy drop, I wonder if it is worth adding the uniform constraint at the cost of such a model accuracy loss.
- Experiments only compare with the kernel pruning methods and lack comparisons with other block pruning methods, e.g. PatDNN [1] and 1xN pruning [2].
- This paper only focuses on CPU, but GPU on mobile is also important. For example, the implementation of the algorithms related to autonomous driving currently heavily relies on Orin GPUs. It is better to analyze the efficiency of this method on GPU, and the uniform pruning method should achieve better speedups.

[1] Wei Niu, et al. PatDNN: Achieving Real-Time DNN Execution on Mobile Devices with Pattern-based Weight Pruning, ASPLOS, 2020.
[2] Mingbao Lin, et al. 1xN Pattern for Pruning Convolutional Neural Networks, IEEE Transactions on Pattern Analysis and Machine Intelligence, 2022.

**Questions:**

- If use the same data augmentation and hyper-parameters to train the dense model with the same 250 epochs, what’s the dense model accuracy? Because hyper-parameters and data augmentation are different from other methods, the comparison is unfair. If you can compare the accuracy loss of different methods compared with their dense baselines, it may better demonstrate the effectiveness of SUBP.
- Is this method effective on vision transformers? Transformer models have been widely used in various computer vision tasks.
- Does the regrow method apply to the experiments in Table 1, including the l1 norm and BPAR?

**Limitations:**

The author discussed the limitation of the method and the societal impacts. Other limitations of my concern can refer to the weaknesses.

---

> ### Author Rebuttal · Authors · 2023-08-06
>
> Many thanks for your responsible reviews that will help us improve the manuscript. Please see the following answers to your questions.
>
> **Q1:** Author should use the same hyperparameters for fair comparisons, as well as dense model baselines for other methods to better demonstrate the effectiveness of the proposed methods.
>
> **A1:** Your suggestion is valuable and we are aware of it. We trained the dense model of **ResNet18 for 250 epochs**, the top-1 and top-5 accuracy are **71.9% and 90.3%** respectively, which brings 2.1% and 1.2% improvement to the torchvision baseline (90 epochs).
>
> Admittedly, increasing the epoch does promote better model fitting and thus better results. Methods based on pre-trained pruning usually require a large number of epochs to fine-tune the pruning model, thus achieving "lossless" pruning of the model. From a certain point of view, it is also a way to compare the fully trained pruned model with the insufficiently trained dense model, which is also unfair to a certain extent.
>
> Therefore, in this paper our main aim is to propose a method that can be trained from scratch to obtain a pruned model. We have chosen epochs that are similar to those of recent methods, in the hope of achieving some degree of fair comparison.
>
> **A2:** Does the method the authors work on Vision Transformer？
>
> **Q2:** Yes, our proposed method can be applied to Vision Transformer. On the one hand, we can equate the densely computed Dense layer in Transformer to a 1x1 convolutional layer, and understand the acceleration based on the acceleration principle of 1x1 convolution. On the other hand, the sparse model of 1xN block pruning is proposed for matrix multiplication, and according to [1], the computation of the Dense layer should have higher computational density than the convolutional layer, so we believe that 1xN block pruning can achieve better results in ViT.
>
> **Q3:** Whether the regrow method  is applied to results in Table1.
>
> **A3:** No. The main purpose of our Table 1 is to show that BPAR-based 1xN block pruning is superior to L1-norm . Thus our results in Table 1 are based on the pre-training-pruning-fine-tuning pruning model, using the same hyper-parameters in [2].
>
> **Q4:** The authors are encouraged to conduct experiments on other tasks.
>
> **A4:** Following your advice, we further conduct experiments on two other tasks including object detection and instance segmentation. Below displays the experimental results in which the proposed SUBP performs well on various vision tasks.
>
> **Object detection results on COCO:**
>
> | **Model**         | **Block Size** | **mAP** |
> | ----------------- | -------------- | ------- |
> | F-RCNN-R50 (4.1G) | -              | 37.4    |
> | F-RCNN-R50 (2G)   | 1x32           | 38.4    |
> | F-RCNN-R50 (2G)   | 1x16           | 38.5    |
> | F-RCNN-R50 (1G)   | 1x32           | 37.1    |
> | F-RCNN-R50 (1G)   | 1x16           | 37.3    |
>
> **Instance segmentation results on COCO:**
>
> | **Model**         | **Block Size** | **Box mAP** | **Mask mAP** |
> | ----------------- | -------------- | ----------- | ------------ |
> | F-RCNN-R50 (4.1G) | -              | 38.2        | 34.7         |
> | F-RCNN-R50 (2G)   | 1x32           | 39.2        | 35.4         |
> | F-RCNN-R50 (2G)   | 1x16           | 39.4        | 35.5         |
> | F-RCNN-R50 (1G)   | 1x32           | 37.4        | 33.8         |
> | F-RCNN-R50 (1G)   | 1x16           | 37.5        | 33.8         |
>
> **Q5:** Authors should make fuller comparisons in their papers.
>
> **A5:** This is our problem and we should place some of the 1xN results from Table 1 in Table 2.  The results are shown as following:
>
> | Block size | Method | Network     | FLOPs | Top-1 accuracy | Epochs  |
> | ---------- | ------ | ----------- | ----- | -------------- | ------- |
> | 32         | 1xN    | ResNet50    | 2.3G  | 76.0%          | 100+180 |
> | 32         | SUBP   | ResNet50    | 2.0G  | 77.4%          | 250     |
> | 32         | 1xN    | MobileNetV1 | 300M  | 69.6%          | 100+180 |
> | 32         | SUBP   | MobileNetV1 | 279M  | 71.1%          | 250     |
> | 16         | 1xN    | ResNet50    | 2.3G  | 76.3%          | 100+180 |
> | 16         | SUBP   | ResNet50    | 2.0G  | 77.6%          | 250     |
> | 16         | 1xN    | MobileNetV1 | 300M  | 69.6%          | 100+180 |
> | 16         | SUBP   | MobileNetV1 | 279M  | 70.8%          | 250     |
>
> **Q6:** 1xN block pruning on GPU.
>
> **A6:**  Recently we find that there have been related papers for accelerated experiments on GPUs for 1xN block pruning. According to [3], 1xN block pruning can achieve significant speedups at lower sparsity ratios on general purpose GPUs. Therefore, we believe that 1xN has the same potential for GPUs.
>
> [1] Ding, Xiaohan, et al. Repmlpnet: Hierarchical vision mlp with re-parameterized locality. *Proceedings of the IEEE/CVF Conference on Computer Vision and Pattern Recognition*. 2022.
>
> [2] Mingbao Lin, et al. 1xN Pattern for Pruning Convolutional Neural Networks, IEEE Transactions on Pattern Analysis and Machine Intelligence, 2022.
>
> [3] Tan, Yijun, et al. Accelerating Sparse Convolution with Column Vector-Wise Sparsity. *Advances in Neural Information Processing Systems* 35 (2022).

---

### Official Review · Reviewer_Njjh · 2023-07-10

**Soundness:** 3 good
**Presentation:** 2 fair
**Contribution:** 3 good
**Rating:** 6
**Confidence:** 4

**Summary:**

Structural weight pruning methods have gain popularity due to the hardware friendly pattern. This work improves 1xN pruning with prune-and-grow and uniform block pruning. Evaluation shows outperforming accuracy results at similar levels of FLOPs than prior work.

**Strengths:**

- This paper presents a new contribution on block-wise pruning with soft (prune-and-grow) in channel-wise uniformly.
- The evaluation shows potential significance of the proposed block pruning method in achieve accuracy with reduced FLOPs.

**Weaknesses:**

- The paper could be improve in clarity and sufficient implemantion details to help readability.

**Questions:**

- Regarding the key design idea of uniform block pruning, what is being done in Section 3.1 and how different compared with prior work of block pruning? Figure 2 could be modified to better illustrate and/or an algorithm description of the steps. Is it every output channel has the same number of pruned 1xN blocks?
- In Section 3.2, block pruning vis angular redundancy could be better positioned in the overall pruning.
- What are the sparsity ratios used in Table 2? While showing FLOPs is informative, it is not the end product, and information on the sparsity ratio could help evaluation the proposed method.

**Limitations:**

Not adequate. From the evaluation, the proposed method, among many other pruning work, significantly increase the training epochs. The negative impacts from more training costs need to be discussed.

---

> ### Author Rebuttal · Authors · 2023-08-06
>
> Thanks for your kind comments. We sincerely wish our response can well address your concerns so as to raise the rating score.
>
> **Q1:** The author explain how it differs from previous block pruning. The author should add an algorithm to describe of the steps better.
>
> **A1:** More specifically. According to [1] and [2], 1xN block pruning implies that **N consecutive output channels have the same sparse pattern.** In Section 3.1, we first introduced the basic concept of 1xN block pruning. However, we find that previous work has only been constrained by Eq.(1), which may result in different pruning rates among different otuput channels and thus unbalanced computational loads across threads. Thus we optimize Eq.(1) for the multi-threads scenario to Eq.(2).
> **For example**, if we assume a Conv2D with a weight (out_channel, in_channel, kernel_size, kernel_size) if (6, 4, 1, 1)，N=2，prune_rate=0.5. We claim 1 means preserved and 0 means pruned. Eq(1) may result in
>
> $$\begin{bmatrix}
>  1 & 0 & 1 & 1 \\\\
>  1 & 0 & 1 & 1 \\\\
>  0 & 1 & 0 & 0 \\\\
>  0 & 1 & 0 & 0 \\\\
>  1 & 1 & 0 & 0 \\\\
>  1 & 1 & 0 & 0
> \end{bmatrix}$$
>
> According to the shortest plank effect, the speed at which the network runs is limited by the first and second output channels. Eq. (2) constraints the sparsity of each output channel to keep the same, we may get
>
> $$\begin{bmatrix}
>  1 & 0 & 0 & 1 \\\\
>  1 & 0 & 0 & 1 \\\\
>  0 & 1 & 1 & 0 \\\\
>  0 & 1 & 1 & 0 \\\\
>  1 & 1 & 0 & 0 \\\\
>  1 & 1 & 0 & 0
> \end{bmatrix}$$
>
> In this case, each output channel has the same computational load, favouring multi-threaded scenarios running.
> To better illustrate the pruning process, we show the relevant **algorithm** description in the **Appendix.B.**
>
> **Q2:** The author could positione the block pruning vis angular redundancy better in the overall pruning.
>
> **A2:** Thank you for your careful reading and we will revise this section appropriately to better present the block pruning vis angular redundancy if our paper is accepted.
>
> **Q3:** The authors only report the training FLOPs, the authors are encouraged to report the sparsity ratio to help evaluation the proposed method.
>
> **A3:**  We appreciate this valuable comment, which helps us to further improve the quality of our paper. The **sparsity ratios (SR)** for Table 2 will be added in the final version if the paper is accepted (the table can not be shown fully for characters limitation). We note that our method can achieve better results in terms of sparsity ratio and accuracy. An intutive explanation is that previous methods with **different sparsity ratios (SR)** between layers tend to pruning the layers that are shallow and these parameters tend to have more computations.  For example, when the Conv2D's weight shape (in_channel, out_channel, kernel_size, kernel_size) is (32, 64, 3, 3), it will cause more FLOPs when input size is [1, 32, 56, 56] than [1, 32, 28, 28]. **In our SUBP, we prune all layers with the same sparsity ratio.**
>
> | Method        | PT | SR    | FLOPs | Top-1 | Epochs |
> |---------------|----|-------|-------|-------|--------|
> | ResNet-18     |    |       |       |       |        |
> |    PFP        | √  | 43.8% | 1.27G | 67.4% | 270    |
> |    SCOP       | √  | 39.3% | 1.10G | 69.2% | 230    |
> |    SFP        | √  | 47.6% | 1.04G | 67.1% | 200    |
> |    FPGM       | √  | 47.6% | 1.04G | 68.4% | 200    |
> |    DMCP       | ×  | 17.6% | 1.04G | 69.0% | 150    |
> |    CHEX       | ×  | 38.7% | 1.03G | 69.6% | 250    |
> |    SUBP(1x16) | ×  | 44.1% | 1.03G | 69.9% | 250    |
> |    SUBP(1x32) | ×  | 44.1% | 1.03G | 69.7% | 250    |
> | ResNet-34     |    |       |       |       |        |
> |    Taylor     | √  | 21.1% | 2.8G  | 72.8% | -      |
> |    SFP        | √  | 49.1% | 2.2G  | 71.8% | 200    |
> |    FPGM       | √  | 49.1% | 2.2G  | 72.5% | 200    |
> |    GFS        | √  | 32.5% | 2.1G  | 72.9% | 240    |
> |    DMC        | √  | -     | 2.1G  | 72.6% | 490    |
> |    NPPM       | √  | -     | 2.1G  | 73.0% | 390    |
> |    SCOP       | √  | 45.6% | 2.0G  | 72.6% | 230    |
> |    CafeNet    | ×  | 21.1% | 1.8G  | 73.1% | 300    |
> |    CHEX       | ×  | 29.2% | 2.0G  | 73.5% | 250    |
> |    SUBP(1x16) | ×  | 43.8% | 2.0G  | 73.7% | 250    |
> |    SUBP(1x32) | ×  | 43.8% | 2.0G  | 73.6% | 250    |
>
> **Q4:** The author should discuss the negative impact from the increasing training cost.
>
> **A4:**  To be honest, our method does increase the training overhead compared to other methods, a process that increases energy consumption and greenhouse gas emissions. Therefore, it is also our future duty to investigate how to improve the convergence speed of the model with limited training resources. If we look at it from a different perspective, in terms of the entire model lifecycle, the gains from these additional training overheads are acceptable to some extent if they can be covered at the time of model deployment.
>
> [1] Elsen, Erich, et al. Fast sparse convnets. *Proceedings of the IEEE/CVF conference on computer vision and pattern recognition*. 2020.
>
> [2] Mingbao Lin, et al. 1xN Pattern for Pruning Convolutional Neural Networks, IEEE Transactions on Pattern Analysis and Machine Intelligence, 2022.

---

### Decision · Program_Chairs · 2023-09-21

**Decision:**

Accept (poster)

**Comment:**

This work improves 1xN pruning with prune-and-grow and uniform block pruning, which can be used for training from scratch to reduce the training cost and prunes the weights in a uniform manner to handle the load imbalance problem. During pruning, the method uses block angular redundancy as the criterion to discard the redundant blocks and allows pruned blocks to regrow to the network based on the importance sampling to search for a better sparse pattern. Experiments show that the SUBP method can achieve accuracy improvements compared with state-of-the-art methods with similar FLOPs. The method adopts uniform pruning to handle the load imbalance problem and shows the real speedup on the CPU platform to demonstrate the actual acceleration.


The paper received scores 3, 5, 6, 6 after the rebuttal. Additional experimental results were added during the rebuttal, which cover more model architectures and downstream tasks for better demonstrating the its effectivness and solved most reviewers' concerns. The main criticism of the score-3 comment focuses on the "marginal improvement over CHEX" and the “limited applications in today's GPU”. Considering the improvement from DMCP to CHEX (60.90% to 69.6%), the improvement of the proposed method (69.6%->70.2%/70.4%) is reasonable. The deployment to actual GPUs might be a challenge for the current compilers. But the GPU makers might contribute to this direction if they find the 1xN pruning methods promising. In addition, CPU makers are also pushing forward deep learning models' deployment on CPUs.

The authors are recommended to include the additional results during the rebuttal in the final version.